# NEURAL SUBGRAPH MATCHING

## ABSTRACT

Subgraph matching is the problem of determining the presence of a given query graph in a large target graph. Despite being an NP-complete problem, the subgraph matching problem is crucial in domains ranging from network science and database systems to biochemistry and cognitive science. However, existing techniques based on combinatorial matching and integer programming cannot handle matching problems with both large target and query graphs. Here we propose NeuroMatch, an accurate, efficient, and robust neural approach to subgraph matching. NeuroMatch decomposes query and target graphs into small subgraphs and embeds them using graph neural networks. Trained to capture geometric constraints corresponding to subgraph relations, NeuroMatch then efficiently performs subgraph matching directly in the embedding space. Experiments demonstrate that NeuroMatch is 100x faster than existing combinatorial approaches and 18% more accurate than existing approximate subgraph matching methods.

## 1. INTRODUCTION

Given a query graph, the problem of subgraph isomorphism matching is to determine if a query graph is isomorphic to a subgraph of a large target graph. If the graphs include node and edge features, both the topology as well as the features should be matched.

Subgraph matching is a crucial problem in many biology, social network and knowledge graph applications (Gentner, 1983; Raymond et al., 2002; Yang & Sze, 2007; Dai et al., 2019). For example, in social networks and biomedical network science, researchers investigate important subgraphs by counting them in a given network (Alon et al., 2008). In knowledge graphs, common substructures are extracted by querying them in the larger target graph (Gentner, 1983; Plotnick, 1997).

Traditional approaches make use of combinatorial search algorithms (Cordella et al., 2004; Gallagher, 2006; Ullmann, 1976). However, they do not scale to large problem sizes due to the NP-complete nature of the problem. Existing efforts to scale up subgraph isomorphism (Sun et al., 2012) make use of expensive pre-processing to store locations of many small 2-4 node components, and decompose the queries into these components. Although this allows matching to scale to large target graphs, the size of the query cannot scale to more than a few tens of nodes before decomposing the query becomes a hard problem by itself.

Here we propose NeuroMatch, an efficient neural approach for subgraph matching. The core of NeuroMatch is to decompose the target $G_T$ as well as the query $G_Q$ into many small overlapping graphs and use a Graph Neural Network (GNN) to embed the individual graphs such that we can then quickly determine whether one graph is a subgraph of another.

Our approach works in two stages, an embedding stage and a query stage. At the embedding stage, we decompose the target graph $G_T$ into many sub-networks $G_u$: For every node $u \in G_T$ we extract a $k$-hop sub-network $G_u$ around $u$ and use a GNN to obtain an embedding for $u$, capturing the neighborhood structure of $u$. At the query stage, we compute embedding of every node $q$ in the query graph $G_Q$ based on $q$'s neighborhood. We then compare embeddings of all pairs of nodes $q$ and $u$ to determine whether $G_Q$ is a subgraph of $G_T$.

The key insight that makes NeuroMatch work is to define an embedding space where subgraph relations are preserved. We observe that subgraph relationships induce a partial ordering over subgraphs. This observation inspires the use of geometric set embeddings such as order embeddings (McFee & Lanckriet, 2009),which induce a partial ordering on embeddings with geometric shapes. By ensuring that the partial ordering on embeddings reflects the ordering on subgraphs, we equip our model with

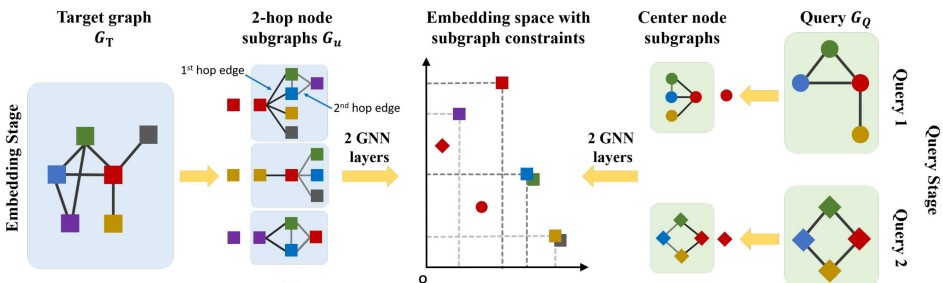

Figure 1: **Overview of NeuroMatch.** We decompose target graph $G_T$ by extracting $k$-hop neighborhood $G_u$ around at every node $u$. We then use a GNN to embed each $G_u$ (left). We refer to $u$ as the center node of $G_u$. We train the GNN to reflect the subgraph relationships: If $G_v$ is a subgraph of $G_u$, then node $v$ should be embedded to the lower-left of $u$. For example, since the 2-hop graph of the violet node is a subgraph of the 2-hop graph of the red node, the embedding of the violet square is to the lower-left of the red square node. At the query stage, we decompose the query $G_Q$ by picking an anchor node $q$ and embed it. From the embedding itself we can quickly determine that Query 1 is a subgraph of the neighborhood around red, blue, and green nodes in target graph because its embedding is to the lower-left of them. Similarly, Query 2 is a subgraph of the purple and red nodes and is thus positioned to the lower-left of both nodes. Notice NeuroMatch avoids expensive combinatorial matching of subgraphs.

a powerful set of inductive biases while greatly simplifying the query process. Our work differs from many previous works (Bai et al., 2019; Li et al., 2019; Xu et al., 2019) that embed graphs into vector spaces, which do not impose geometric structure in the embedding space. In contrast, order embeddings have properties that naturally correspond to many properties of subgraph relationships, such as transitivity, symmetry and closure under intersection. Enforcing the order embedding constraint both leads to a well-structured embedding space and also allows us to efficiently navigate it in order to find subgraphs as well as supergraphs (Fig. 1).

NeuroMatch trains a graph neural network to learn the order embedding, and uses a max-margin loss to ensure that the subgraph relationships are captured. Furthermore, the embedding stage can be conducted *offline*, producing precomputed embeddings for the query stage. The query stage is extremely efficient due to the geometric constraints imposed at training time, and it only requires linear time both in the size of the query and the target graphs. Lastly, NeuroMatch can naturally operate on graphs which include categorical node and edge features, as well as multiple target graphs.

We compare the accuracy and speed of NeuroMatch with state-of-the-art exact and approximate methods for subgraph matching (Cordella et al., 2004; Bonnici et al., 2013) as well as recent neural methods for graph matching, which we adapted to the subgraph matching problem. Experiments show that NeuroMatch runs two orders of magnitude faster than exact combinatorial approaches and can scale to larger query graphs. Compared to neural graph matching methods, NeuroMatch achieves an 18% improvement in AUROC for subgraph matching. Furthermore, we demonstrate the generalization of NeuroMatch, by testing on queries sampled with different sampling strategies, and transferring the model trained on synthetic datasets to make subgraph predictions on real datasets.

## 2. NeuroMatch Architecture

### 2.1. Problem Setup

We first describe the general problem of subgraph matching. Let $G_T = (V_T, E_T)$ be a large *target graph* where we aim to identify the query graph. Let $X_T$ be the associated categorical node features for all nodes in $V$[1]. Let $G_Q = (V_Q, E_Q)$ be a *query graph* with associated node features $X_Q$. The goal of a subgraph matching algorithm is to identify the set of all subgraphs $\mathcal{H} = \{H | H \subseteq G_T\}$ that are isomorphic to $G_Q$, that is, $\exists$ bijection $f : V_H \mapsto V_Q$ such that $(f(v), f(u)) \in E_Q$ iff $(v, u) \in E_H$. Furthermore, we say $G_Q$ is a subgraph of $G_T$ if $\mathcal{H}$ is non-empty. When node and edge features are present, the subgraph isomorphism further requires that the bijection $f$ has to match these features.

---

[1]We consider the case of a single target and query graph, but NeuroMatch applies to any number of target/query graphs. We also assume that the query is connected (otherwise it can be easily split into 2 queries).

---

**Algorithm 1:** NeuroMatch Query Stage

---

**Input:** Target graph $G_T$, graph embeddings $Z_u$ of node $u \in G_T$, and query graph $G_Q$.
**Output:** Subgraph of $G_T$ that is isomorphic to $G_Q$.
 1: For every node $q \in G_Q$, create $G_q$, and embed its center node $q$.
 2: Compute matching between embeddings $Z_q$ and embeddings $Z_T$ using subgraph prediction function $f(z_q, z_u)$.
 3: Repeat for all $q \in G_Q$, $u \in G_T$; make prediction based on the average score of all $f(z_q, z_u)$.

---

In the literature, subgraph matching commonly refers to two subproblems: node-induced matching and edge-induced matching. In node-induced matching, the set of possible subgraphs of $G_T$ are restricted to graphs $H = (V_H, E_H)$ such that $V_H \subseteq V_T$ and $E_H = \{(u,v)|u, v \in V_H, (u,v) \in E_T\}$. Edge-induced matching, in contrast, restricts possible subgraphs by $E_H \subseteq E_T$, and contains all nodes that are incident to edges in $E_H$. To demonstrate, here we consider the more general edge-induced matching, although NeuroMatch can be applied to both.

In this paper, we investigate the following *decision* problems of subgraph matching.

**Problem 1.** *Matching query to datasets.* *Given a target graph $G_T$ and a query $G_Q$, predict if $G_Q$ is isomorphic to a subgraph of $G_T$.*

We use neural model to decompose Problem 1 and solve (with certain accuracy) the following neighborhood matching subproblem.

**Problem 2.** *Matching neighborhoods.* *Given a neighborhood $G_u$ around node $u$ and query $G_Q$ anchored at node $q$, make binary prediction of whether $G_q$ is a subgraph of $G_u$ where node $q$ corresponds to $u$.*

Here we define an *anchor* node $q \in G_Q$, and predict existence of subgraph isomorphism mapping that also maps $q$ to $u$. At prediction time, similar to (Bai et al., 2018), we compute the alignment score that measures how likely $G_Q$ anchored at $q$ is a subgraph of $G_u$, for all $q \in G_Q$ and $u \in G_u$, and aggregate the scores to make the final prediction to Problem 1.

## 2.2. OVERVIEW OF NEUROMATCH

NeuroMatch adopts a two-stage process: *embedding stage* where $G_T$ is decomposed into many small overlapping graphs and each graph is embedded. And the *query stage* where query graph is compared to the target graph directly in the embedding space so no expensive combinatorial search is required.

**Embedding stage**. In the embedding stage, NeuroMatch decomposes target graph $G_T$ into many small overlapping neighborhoods $G_u$ and uses a graph neural network to embed them. For every node $u$ in $G_T$, we extract the $k$-hop neighborhood of $u$, $G_u$ (Figure 1). GNN then maps node $u$ (that is, the structure of its network neighborhood $G_u$) into an embedding $z_u$.

Note a subtle but an important point: By using a $k$-layer GNN to embed node $u$, we are essentially embedding/capturing the $k$-hop network neighborhood structure $G_u$ around the center node $u$. Thus, embedding $u$ is equivalent to embedding $G_u$ (a $k$-hop subgraph centered at node $u$), and by comparing embeddings of two nodes $u$ and $v$, we are essentially comparing the structure of subgraphs $G_u, G_v$.

**Query stage (Alg. 1)**. The goal of the query stage is to determine whether $G_Q$ is a subgraph of $G_T$ and identify the mapping of nodes of $G_Q$ to nodes of $G_T$. However, rather than directly solving this problem, we develop a fast routine to determine whether $G_q$ is a subgraph of $G_u$: We design a *subgraph prediction function* $f(z_q, z_u)$ that *predicts* whether the $G_Q$ anchored at $q \in G_Q$ is a subgraph of the $k$-hop neighborhood of node $u \in G_T$, which implies that $q$ corresponds to $u$ in the subgraph isomorphism mapping by Problem 2. We thus formulate the subgraph matching problem as a node-level task by using $f(z_q, z_u)$ to predict the set of nodes $v$ that can be matched to node $q$ (that is, find a set of graphs $G_u$ that are super-graphs of $G_q$). To determine wither $G_Q$ is a subgraph of $G_T$, we then aggregate the alignment matrix consisting of $f(z_q, z_u)$ for all $q \in G_Q$ and $u \in G_T$ to make the binary prediction for the decision problem of subgraph matching.

**Practical considerations and design choices**. The choice of the number of layers, $k$, depends on the size of the query graphs. We assume $k$ is at least the diameter of the query graph, to allow the information of all nodes to be propagated to the anchor node in the query. In experiments, we observe that inference via voting can consistently reach peak performance for $k = 10$, due to the small-world property of many real-world graphs.

NeuroMatch is flexible in terms of the GNN model used for the embedding step. We adopt a variant of GIN (Xu et al., 2018) incorporating skip layers to encode the query graphs and the neighborhoods, which shows performance advantages. Although GIN showed limitation in expressive power beyond WL test, our GNN additionally uses a feature to distinguish anchor nodes, which results in higher expressive power in distinguishing d-regular graphs, beyond WL test (see Limitation Section and Appendix I).

## 2.3. SUBGRAPH PREDICTION FUNCTION $f(z_q, z_u)$

Given the target graph node embeddings $z_u$ and the center node $q \in G_Q$, the subgraph prediction function decides if $u \in G_T$ has a $k$-hop neighborhood that is subgraph isomorphic to $q$'s $k$-hop neighborhood in $G_Q$. The key is that subgraph prediction function makes this decision based only on the embeddings $z_q$ and $z_u$ of nodes $q$ and $u$ (Figure 1).

**Capturing subgraph relations in the embedding space**. We enforce the embedding geometry to directly capture subgaph relations. This approach has the additional benefit of ensuring that the subgraph predictions have negligible cost at the query stage, since we can just compare the coordinates of two node embeddings. In particular, NeuroMatch satisfies the following properties for subgraph relations (Refer to Appendix A for proofs of the properties):

- *Transitivity:* If $G_1$ is a subgraph of $G_2$ and $G_2$ is a subgraph of $G_3$, then $G_1$ is a subgraph of $G_3$.
- *Anti-symmetry:* If $G_1$ is subgraph of $G_2$, $G_2$ is a subgraph of $G_1$ iff they are isomorphic.
- *Intersection set:* The intersection of the set of $G_1$'s subgraphs and the set of $G_2$'s subgraphs contains all common subgraphs of $G_1$ and $G_2$.
- *Non-trivial intersection:* The intersection of any two graphs contains at least the trivial graph.

We use the notion of set embeddings (McFee & Lanckriet, 2009) to capture these inductive biases. Common examples include order embeddings and box embeddings. In contrast to Euclidean point embeddings, set embeddings enjoy properties that correspond naturally to the subgraph relationships.

**Subgraph prediction function**. The idea of order embeddings is illustrated in Figure 1. Order embeddings ensure that the subgraph relations are properly reflected in the embedding space: if $G_q$ is a subgraph of $G_u$, then the embedding $z_q$ of node $q$ has to be to the "lower-left" of $u$'s embedding $z_u$:

$$z_q[i] \leq z_u[i] \forall_{i=1}^{D} \quad \text{iff} \quad G_q \subseteq G_u \tag{1}$$

where $D$ is the embedding dimension. We thus train the GNN that produces the embeddings using the max margin loss:

$$\mathcal{L}(z_q, z_u) = \sum_{(z_q, z_u) \in P} E(z_q, z_u) + \sum_{(z_q, z_u) \in N} \max\{0, \alpha - E(z_q, z_u)\}, \text{where} \tag{2}$$

$$E(z_q, z_u) = ||\max\{0, z_q - z_u\}||_2^2 \tag{3}$$

Here $P$ denotes the set of positive examples in minibatch where the neighborhood of $q$ is a subgraph of neighborhood of $u$, and $N$ denotes the set of negative examples. A violation of the subgraph constraint happens when in any dimension $i$, $z_q[i] > z_u[i]$, and $E(z_q, z_u)$ represents its magnitude. For positive examples $P$, $E(z_q, z_u)$ is minimized when all the elements in the query node embedding $z_q$ are less than the corresponding elements in target node embedding $z_u$. For negative pairs $(z_q, z_u)$ the amount of violation $E(z_q, z_u)$ should be at least $\alpha$, in order to have zero loss.

We further use a threshold $t$ on the violation $E(z_q, z_u)$ to make decision of whether the query is a subgraph of the target. The subgraph prediction function $f$ is defined as:

$$f(z_q, z_u) = \begin{cases} 1 & \text{iff } E(z_q, z_u) < t \\ 0 & \text{otherwise} \end{cases} \tag{4}$$

## 2.4. MATCHING NODES VIA VOTING

At the query time, our goal is to predict if query node $q \in G_Q$ and target node $u \in G_T$ have subgraph-isomorphic $k$-hop neighborhoods $G_q$ and $G_u$ (Problem 2). A simple solution is to use the subgraph prediction function $f(z_q, z_u)$ to predict the subgraph relationship between $G_q$ and $G_u$.

**Matching via voting**. We further propose a voting method that improves the accuracy of matching a pair of anchor nodes based on their neighboring nodes. Our insight is that matching a pair of anchor

nodes imposes constraints on the neighborhood structure of the pair. Suppose we want to predict if node $q \in G_Q$ and node $u \in G_T$ match. We have (proof in Appendix C):

**Observation 1.** *Let $\mathcal{N}^{(k)}(u)$ denote the $k$-hop network neighborhood of node $u$. Then, if $q \in G_q$ and node $u \in G_u$ match, then for all nodes $i \in N^{(k)}(q)$, $\exists$ node $j \in \mathcal{N}^{(l)}(u), l \leq k$ such that node $i$ and node $j$ match.*

Based on this observation, we propose a voting-based inference method. Suppose that node $q \in G_Q$ matches node $u \in G_T$. We check if all neighbors of node $q$ satisfy Observation 1, *i.e.* each neighbor of $q$ has a match to neighbor of $u$, as summarized in Appendix Algorithm 2.

## 2.5. TRAINING NEUROMATCH

The training of subgraph matching consists of the following component: (1) Sample training query $G_Q$ from target graph $G_T$. (2) Sample node $q$ and neighborhood $G_q$ in $G_Q$ and find $q$'s corresponding node $u$ and its $G_u \subseteq G_T$. (3) Generate negative example $w$ and its $G_w \subseteq G_T$. (4) Compute node embeddings for $q$, $u$, $w$ with GNN, and the loss in Equation 2 for backprop. We now detail the following components in this training process.

**Training data**. To achieve high generalization performance on unseen queries, we train the network with randomly generated query graphs. We sample a positive pair, we sample $G_u \in G_T$, and $G_q \in G_u$. To sample $G_u$, we first selecting a node $u \in G_T$, and perform a random breadth-first traversal (BFS) of the graph. The sampler traverse each edge in BFS with a fixed probability. We then sample $G_q$ by performing the same random BFS traversal on $G_u$ starting at $u$, and treat $u$ as the anchor in $G_q$, which ensures existence of subgraph isomorphism mapping that maps $q$ to $u$.

Given a positive pair $(G_q, G_u)$, we generate 2 types of negative examples. The first type of negative examples are created by randomly choosing different nodes $u$ and $q$ in $G_T$ and perform random traversal. The second type of negatives are generated by perturbing the query to make it no longer a subgraph of the target graph, which is a more challenging case for the model to distinguish.

**Test data**. To demonstrate generalization, we use 3 different sampling strategies to generate test queries. Aside from the mentioned random BFS traversal, we further use the random walk sampling by performing random walk with restart at $u$, and the degree-weighted sampling strategy used in the motif mining algorithm MFinder (Cho et al., 2013). Experiments demonstrate that NeuroMatch can generalize to test queries with different sampling strategies.

**Curriculum**.
We introduce a curriculum training scheme that improves performance. We first train the model on a small number of easy queries and then train on successively more complex queries with increased batch size. Initially the model is trained with a single 1 hop query. Each time the training performance plateaus, the model samples larger queries. Figure 2 shows examples of queries at each curriculum level. The complexity of queries increases as training proceeds.

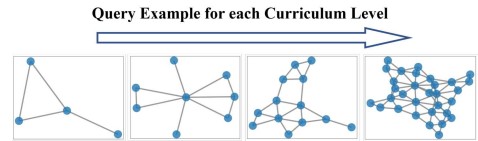
Query Example for each Curriculum Level

Figure 2: Example sampled queries $G_Q$ at each level of the curriculum in the MSRC_21 dataset. The diameter and number of nodes increase as curriculum level advances.

## 2.6. RUNTIME COMPLEXITY

The embedding stage uses GNNs to train embeddings to obey the subgraph constraint. Its complexity is $O(K(|E_T| + |E_Q|))$, where $K$ is the number of GNN layers. In the query stage, to solve Problem 1 we need to compute a total of $O(|V_T||V_Q|)$ scores. The quadratic time complexity allows NeuroMatch to scale to larger datasets, whereas the complexity of the exact methods grow exponentially with size.

In many use cases, the target graphs are available in advance, but we need to solve for new incoming unseen queries. Prior to inference time, the embeddings for all nodes in the target graph can be pre-computed with complexity $O(K|E_T|)$. For a new query, its node embeddings can be computed in $O(K|E_Q|)$ time, which is much faster since queries are smaller. With order embedding, we do not need additional neural network modules at query stage and simply compute the order relations between query node embeddings and the pre-computed node embeddings in the target graph.

## 3. EXPERIMENTS

To investigate the effectiveness of NeuroMatch, we compare its runtime and performance with a range of existing popular subgraph matching methods. We evaluate performance on synthetic datasets

| Dataset | SYNTHETIC | COX2 | DD | MSRC_21 | FIRSTMMDB | PPI | WORDNET18 |
|---|---|---|---|---|---|---|---|
| GMNN (Xu et al., 2019) | $73.6 \pm 1.1$ | $75.9 \pm 0.8$ | $80.6 \pm 1.5$ | $82.5 \pm 1.7$ | $81.5 \pm 2.9$ | $72.0 \pm 1.9$ | $80.3 \pm 2.0$ |
| RDGCN (Wu et al., 2019) | $79.5 \pm 1.2$ | $80.1 \pm 0.4$ | $81.3 \pm 1.2$ | $81.9 \pm 1.9$ | $82.4 \pm 3.4$ | $76.8 \pm 2.2$ | $79.6 \pm 2.5$ |
| NO CURRICULUM | $82.4 \pm 0.6$ | $95.0 \pm 1.6$ | $96.7 \pm 2.1$ | $89.2 \pm 2.0$ | $87.2 \pm 6.8$ | $82.6 \pm 1.7$ | $81.4 \pm 2.2$ |
| NM-MLP | $88.7 \pm 0.5$ | $95.4 \pm 1.6$ | $97.1 \pm 0.3$ | $93.5 \pm 1.0$ | $92.9 \pm 4.3$ | $85.5 \pm 1.4$ | $86.3 \pm 0.9$ |
| NM-NTN | $89.1 \pm 1.9$ | $89.3 \pm 0.9$ | $96.4 \pm 1.4$ | $94.7 \pm 3.2$ | $89.6 \pm 1.1$ | $85.7 \pm 2.4$ | $85.0 \pm 1.1$ |
| NM-BOX | $84.5 \pm 2.1$ | $88.5 \pm 1.2$ | $91.4 \pm 0.5$ | $90.8 \pm 1.4$ | $93.1 \pm 1.7$ | $77.4 \pm 3.1$ | $82.7 \pm 2.5$ |
| NEUROMATCH | $\mathbf{93.5} \pm 1.1$ | $\mathbf{97.2} \pm 0.4$ | $\mathbf{97.9} \pm 1.3$ | $\mathbf{96.1} \pm 0.2$ | $\mathbf{95.5} \pm 2.1$ | $\mathbf{89.9} \pm 1.9$ | $\mathbf{89.3} \pm 2.4$ |
| % IMPROVEMENT | 4.9 | 1.9 | 0.8 | 1.5 | 2.6 | 4.9 | 3.4 |

Table 1: Given a neighborhood $G_u$ of $u$ and query $G_Q$ containing $q$, make binary prediction of whether $G_u$ is a subgraph of $G_u$ where node $q$ corresponds to $u$. We report AUROC (unit 0.01). NeuroMatch performs the best with median AUROC 95.5, 20% higher than the neural baselines.

to probe data efficiency and generalization ability, as well as a variety of real-world datasets spanning many fields to evaluate whether the model can be adapted to real-world graph structures.

### 3.1. DATASETS AND BASELINES

**Synthetic dataset**. We use a synthetic dataset including Erdős-Rényi (ER) random graphs (Erdős & Rényi, 1960) and extended Barabasi graphs (Albert & Barabási, 2000). At test time, we evaluate on test query graphs that were not seen during training. See Appendix E for dataset details, where we also show experiments to transfer the learned model to unseen real dataset without fine-tuning.

**Real-world datasets**. We use a variety of real-world datasets from different domains. We evaluate on graph benchmarks in chemistry (COX2), biology (Enzymes, DD, PPI networks), image processing (MSRC_21), point cloud (FIRSTMMDB), and knowledge graph (WORDNET18). We do not include node features for PPI networks since the goal is to match various protein interaction patterns without considering the identity of proteins. WORDNET18 contains no node features, but we use its edge types information in matching. For all other datasets, we require that the matching takes categorical features of nodes into account. Refer to the Appendix for statistics of all datasets.

**Baselines**. We first consider popular existing combinatorial approaches. We adopt the most commonly used efficient methods: the VF2 (Cordella et al., 2004) and the RI algorithm (Bonnici et al., 2013). We further consider popular approximate matching algorithms FastPFP (Lu et al., 2012) and IsoRankN (Liao et al., 2009), and compare with neural approaches in terms of accuracy and runtime.

Recent development of GNNs has not been applied to subgraph matching. We therefore adapt two recent state-of-the-art methods for graph matching, Graph Matching Neural Networks (GMNN) (Xu et al., 2019) and RDGCN (Wu et al., 2019), by changing their objective from predicting whether two graphs have a match to predicting the subgraph relationship. Both methods are computationally more expensive than NeuroMatch due to cross-graph attention between nodes.

**Training details**. We use the epoch with the best validation result for testing. See Appendix D for hardware and hyperparameter configurations.

### 3.2. RESULTS

**(1) Matching individual node network neighborhoods (Problem 2)**. Table 1 summarizes the AUROC results for predicting subgraph relation for Problem 2: is node $q$'s $k$-hop neighborhood $G_q$ a subgraph of $u$'s neighborhood $G_u$. This is a subroutine to determine is a query is present in a large target graph. The number of pairs $G_q, G_u$ with positive labels is equal to the number of pairs with negative labels. We observe that NeuroMatch with order embeddings obtains, on average, a 20% improvement over neural baselines. This benefit is a result of avoiding the loss of information when pooling node embeddings and a better inductive bias stemming from order embeddings.

**(2) Ablation studies**. Although learning subgraph matching has not been extensively studied, we explore alternatives to components of NeuroMatch. We compare with the following variants:

- NO CURRICULUM: Same as NEUROMATCH but with no curriculum training scheme.
- NM-MLP: uses MLP and cross entropy to replace the order embedding loss.
- NM-NTN: uses Neural Tensor Network (Socher et al., 2013) and cross entropy to replace order embedding loss.
- NM-BOX: uses box embedding loss (Vilnis et al., 2018) to replace the order embedding loss.

| Dataset | COX2 | DD | MSRC_21 | FIRSTMMDB | ENZYMES | SYNTHETIC | Avg Runtime |
|---|---|---|---|---|---|---|---|
| ISORANKN | $72.1 \pm 2.5$ | $61.2 \pm 1.3$ | $67.0 \pm 2.0$ | $77.0 \pm 2.3$ | $50.4 \pm 1.4$ | $62.7 \pm 3.4$ | $1.45 \pm 0.04$ |
| FASTPFP | $63.2 \pm 3.8$ | $72.9 \pm 1.1$ | $83.5 \pm 1.5$ | $83.0 \pm 1.5$ | $76.6 \pm 1.9$ | $77.0 \pm 2.0$ | $0.56 \pm 0.01$ |
| NM-MLP | $73.8 \pm 3.7$ | $87.8 \pm 1.5$ | $74.2 \pm 1.0$ | $88.9 \pm 0.9$ | $87.9 \pm 1.0$ | $\mathbf{92.1} \pm 0.5$ | $1.29 \pm 0.10$ |
| NEUROMATCH | $\mathbf{89.9} \pm 1.1$ | $\mathbf{95.7} \pm 0.4$ | $\mathbf{84.5} \pm 1.5$ | $\mathbf{91.9} \pm 1.0$ | $\mathbf{92.9} \pm 1.2$ | $75.2 \pm 1.8$ | $0.90 \pm 0.09$ |

Table 2: Given a query $G_Q$ and a target graph $G_T$ from a dataset, make binary prediction for whether $G_Q$ is a subgraph of $G_T$ (the decision problem of subgraph isomorphism), in AUROC (unit: 0.01).

As shown in Table 1, box embeddings cannot guarantee intersection, *i.e.* common subgraphs, between two graphs, while variable sizes of the target graph makes neural tensor network (NTN) variant hard to learn. NeuroMatch outperforms all the variants.

We additionally observe that the learning curriculum is crucial to the performance of learning the subgraph relationships. The use of the curriculum increases the performance by an average of 6%, while significantly reducing the performance variance and increasing the convergence speed. This benefit is due to the compositional nature of the subgraph matching task.

**3) Matching query to target graph (Problem 1)**. Given a target $G_T$, we randomly sample a query $G_Q$ centered at $q$. The goal is to answer the decision problem of whether $G_Q$ is a subgraph of $G_T$. Unlike the previous tasks, it requires prediction of subgraph relations between $G_Q$ and neighborhoods $G_u$ for all $u \in G_T$. We perform the tasks by traversing over all nodes in query graph, and all nodes in target graph as anchor nodes, and outputs an alignment matrix $\mathcal{A}$ of dimension $|V_T|$-by-$|V_Q|$, where $\mathcal{A}_{i,j}$ denotes the matching score $f(z_i, z_j)$, as illustrated in Algorithm 1. The performance trend of Table 1 also holds here in Problem 1. We further compare NeuroMatch with high-performing hueristic methods, FastPFP and IsoRankN, and show an average of 18.4% improvement in AUROC over all datasets. Appendix D contains additional implementation details.

Additionally, we make the task harder by sampling test queries with a different sampling strategy. At training time, the query is randomly sampled with the random BFS procedure, whereas at test time the query is randomly sampled using degree-weighted sampling (see Section 2.5).

We further compute the statistics of query graphs and target graphs (in Appendix E). On average across all datasets, the size of query is 51% of the size of the target graphs, indicating that the model is learning the problem of subgraph matching in a data-driven way, rather than learning graph isomorphism, which previous works focus on.

**4) Generalization**. We further conduct experiments to demonstrate the generalization of NeuroMatch.

Firstly, we investigate model generalization to unseen subgraph queries sampled from different distributions. We consider 3 sampling strategies: random BFS, degree-weighted sampling and random walk sampling (see Section 2.5). Table 3 shows the performance of NeuroMatch when trained with examples sampled with one strategy (rows), and tested with examples sampled with another strategy (columns). We observe that NeuroMatch can generalize to queries generated with different sampling strategies, without much performance change. Among strategies considered, random BFS is the most robust sampling strategy for training.

|  | BFS | MFinder | Random Walks |
|---|---|---|---|
| BFS | 98.79 | 98.58 | 98.38 |
| MFinder | 93.09 | 96.34 | 96.07 |
| Random Walks | 95.65 | 97.21 | 97.53 |

Table 3: Generalization to new sampling methods for MSRC dataset. Performance measured in AUROC (unit 0.01).

Secondly, we investigate whether the model is able to generalize to perform matching on pairs of query and target that are from a variety of datasets, while only training on a synthetic dataset. In Appendix F, we similarly find that NeuroMatch is robust to test queries sampled from different real-world datasets.

**Order embedding space analysis**. Figure 3 shows the TSNE embedding of the learned order embedding space. The yellow color points correspond to embeddings of graphs with larger sizes; the purple color points correspond to embeddings of graphs with smaller sizes. Red points are example embeddings for which we also visualize the corresponding graphs. We observe that the order constraints are well-preserved. We further conduct experiment by randomly sampling 2 graphs in the dataset and test their subgraph relationship. NeuroMatch achieves 0.61 average precision, compared to 0.35 with the NM-MLP baseline.

**Comparison with exact methods**.

Although exact methods always achieve the correct answer, they take exponential time in worst case. We run the exact methods VF2 and RI and record the average runtime, using exactly the same test queries and target as in Table 2. If the subgraph matching runs for more than 10 minutes, it is deemed as unsuccessful. We show in Appendix F the runtime comparison showing 100 times speedup with NeuroMatch, and the figure of the success rate of the baselines, which drop below 60% when the query size is more than 30. As query size grows, the runtime of the exact methods grow exponentially, whereas the runtime of NeuroMatch grows linearly. Although VF2 and RI are exact algorithms, NeuroMatch shows the poten-

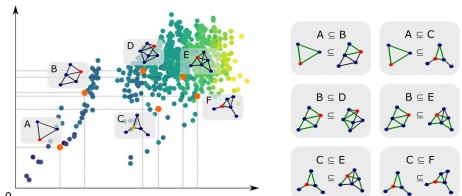

Figure 3: TSNE visualization of order embedding for a subset of subgraphs sampled from the ENZYMES dataset. As seen by examples to the right, the order constraints are well-preserved. Graphs are colored by number of edges.

tial of learning to predict subgraph relationship, in applications requiring high-throughput inference. Additionally, NeuroMatch is also 10 times more efficient than the other baselines such as NM-MLP and GMNN due to its efficient inference using order embedding properties.

## 4. LIMITATIONS

NeuroMatch provides a novel approach to demonstrate the promising potential of GNNs and geometric embeddings to make predictions of subgraph relationships. However, future work is needed in exploring neural approaches to this NP-Complete problem. Previous works (Xu et al., 2018) have identified expressive power limitations of GNNs in terms of the WL graph isomorphism test. In NeuroMatch, we alleviate the limitation by distinguishing between the anchor node via node features (illustrated in Appendix H). Since NeuroMatch does not explicitly rely on a GNN backbone, future work on more expressive GNNs can be directly applied to NeuroMatch. We hope that NeuroMatch opens a new direction in investigating subgraph matching as a potential application and benchmark in graph representation learning.

## 5. RELATED WORK

**Subgraph matching algorithms**. Determining if a query is a subgraph of a target graph requires comparison of their structure and features (Gallagher, 2006). Conventional algorithms (Ullmann, 1976) focus on graph structures only. Other works (Aleman-Meza et al., 2005; Coffman et al., 2004) also consider categorical node features. Our NeuroMatch model can operate under both settings. Approximate solutions to the problem have also been proposed (Christmas et al., 1995; Umeyama, 1988) NeuroMatch is related in a sense that it is an approximate algorithm using machine learning. We further provide detailed comparison with a survey of heuristic methods (Ribeiro et al., 2019).

**Neural graph matching**. Earlier work (Scarselli et al., 2008) has demonstrated the potential of GNN in small-scale subgraph matching, showing advantage of GNN over feed forward neural networks. Recently, graph neural networks (Kipf & Welling, 2017; Hamilton et al., 2017; Xu et al., 2018) have been proposed for graph isomorphism (Bai et al., 2019; Li et al., 2019; Guo et al., 2018) and have achieved state-of-the-art results (Zhang & Lee, 2019; Wang et al., 2019; Xu et al., 2019). However, these methods cannot be directly employed in subgraph isomorphism since there is no one-to-one correspondence between nodes in query and target graphs. We demonstrate that our contributions in using node-based representations, order embedding space can significantly outperform applications of graph matching methods in the subgraph isomorphism setting. Additionally, recent works (Bai et al., 2018; Fey et al., 2020) provide solutions to compute discrete matching correspondences from the neural prediction of isomorphism mapping and are complementary to our work.

## 6. CONCLUSION

In this paper we presented a neural subgraph matching algorithm, NeuroMatch, that uses graph neural networks and geometric embeddings to learn subgraph relationships. We observe that order embeddings are natural fit to model subgraph relationships in embedding space. NeuroMatch outperforms adaptations of existing graph-isomorphism related architectures and show advantages and potentials compared to heuristic algorithms.

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
