# OpenReview forum: "Neural Subgraph Matching"
_ICLR.cc/2021/Conference — Reject_

### Official Review · AnonReviewer1 · 2020-10-27
**Interesting application of GNNs but lacks appropriate background study**

**Rating:** 5
**Confidence:** 3

**Review:**

This paper presents a sub-graph isomorphism method using neural networks and embeddings that are supposed to preserve important topological structure of subgraphs. Sub-graph isomorphism is NP-complete and has fascinated researchers with approximate solutions using heuristics and pre-processing (graph indexes) of many varieties over the years. The approach of using a neural network to learn to perform tasks such as link prediction, label prediction etc., through meaningful node embeddings, is a welcome extension when applied to sub-graph matching.

That being said, the paper skips over large volumes of progress in sub-graph matching work. From indexing methods, analysis of vertex relationships etc., many methods have proven efficient and effective in sub-graph isomorphism ( http://www.vldb.org/pvldb/vol8/p617-ren.pdf , https://dl.acm.org/doi/10.1145/2463676.2465300 , https://dl.acm.org/doi/10.1145/3299869.3319880 to cite a few) . It is only appropriate that a paper on this topic present a reasonable analysis of the prior work. Also, the embedding design decisions could possibly take-away important conclusions from these methods and lastly, compare against solid benchmarks instead of much older and outdated ones.

In conclusion, this work is quite interesting. However, at the moment it is not clear for this reviewer how to put it in perspective with respect to decades of systems and theoretical work on sub-graph isomorphism. A fair empirical evaluation should compare the GNN method to one of the state of the art non-neural methods.

Thanks to the authors for providing a response to the review comments.

---

> ### Author Response · Authors · 2020-11-24
> **Response to Reviewer #1**
>
> We thank the reviewer for acknowledging the novelty of applying graph representation learning to the challenging subgraph matching problem, and for bringing up relevant recent work.
>
> RE: Further compare to algorithmic baselines in related work and experiments.
>
> We emphasize that the main focus of this paper is to explore a new neural approach to this problem and as such, many of the alternative subgraph matching algorithms with heuristics (often highly hand-engineered) are not related in methodology. We include them for completeness, but our contribution in this field is unique. However, we agree that it is important to put our neural approach in the context of prior work and have taken the following steps, in addition to our existing comparisons against algorithmic baselines.
>
> We include discussion of these most recent subgraph matching algorithms in related work. We categorized previous approaches into the category of search space pruning, efficient search space navigation and integer programming, and demonstrated how components of our algorithm (search space representation) fits into the line of work. However, we also point out the innovation of our work, and how it differs in principle to previous approaches. We included these discussions in the revised related work section.
> We run experiments to compare NeuroMatch with the recent DAF method (Han et al 2019) mentioned by the reviewer. We compare DAF against NeuroMatch on runtime matching queries on the DD dataset, and find that only 53% of queries are completed by DAF within the time limit of 20 seconds. In comparison, NeuroMatch handles 100% of queries within the time limit with a tradeoff of small prediction error (acceptable in certain real-world applications such as matching social patterns or filtering promising molecule candidates). We added a detailed description of this experiment to the appendix.
>
>
> ----------------------------------
> ### Search space pruning
> https://journals.plos.org/plosone/article/authors?id=10.1371/journal.pone.0097896
> https://researcher.watson.ibm.com/researcher/files/us-ytian/tale.pdf
> http://www.vldb.org/pvldb/vol8/p617-ren.pdf
> https://dl.acm.org/doi/10.1145/2463676.2465300 https://dl.acm.org/doi/10.1145/3299869.3319880
>
> ### Integer linear program + relaxations
> https://link.springer.com/article/10.1007/s10601-009-9074-3
> https://www.sciencedirect.com/science/article/abs/pii/S0031320312002683
> https://arxiv.org/abs/1207.1114
> https://academic.oup.com/bioinformatics/article/25/12/i253/189039

---

### Official Review · AnonReviewer3 · 2020-10-29
**Subgraph matching by geometric embedding**

**Rating:** 5
**Confidence:** 3

**Review:**

The authors propose neural network schemes for subgraph isomorphism.  Their main subroutine is a subgraph embedding that is trained to encourage the property that if A is a subgraph of B then the embedding emb(B) coordinatewise dominates emb(A).  Given this subroutine, the paper proceeds by producing an embedding of the neighborhood around each vertex of the original graph, then searching for the embeddings of the query subgraph, again centered at each node.  The results of this computation are then used to guess an alignment between the vertices of the query and (a subset of) the vertices of the target graph.

Strengths of the paper:
* The authors propose the study of this neural subgraph matching problem.  I believe we're past the days in which applying deep learning to a new problem is considered a novel contribution in its own right, but it's still an interesting topic to read about.
* The authors generate training data for the system by sampling subgraphs from a given graph.  This has the advantage of tailoring the subgraphs they find to the input distribution, and provides a clean way to structure the training data.  They also do some nice study of alternate sampling approaches at inference, to test sensitivity to the particulars of the data generation scheme.
* The authors propose a particular approach to the embedding learning based on geometric set embeddings that is well chosen for this problem, and seems to perform well in practice.
* The post-inference alignment problem is a way to bring significant knowledge about the problem domain into the solution, after the learned part of the system completes.

Weaknesses of the paper:
* I feel that the runtime isn't well described or analyzed, including in Section F of the supplementary material appendix.  Without the coordinatewise constraint on embeddings, it would be natural to implement the matching of each query subgraph using nearest neighbor search.  However, I didn't see discussion of how to perform the lookups efficiently in the proposed scheme.  Scanning the entire database of target subgraphs doesn't seem ideal.
* I don't have a good sense of whether an 18% improvement from a general graph matching solution to this tailored solution for subgraphs is a significant advance or not.
* I felt that the details of the sampling scheme and the alignment algorithm would result in significant changes to the performance of the system, but I didn't see detailed discussion of these issues.

Questions:
Q1: Could the authors speak a little to efficiently querying for the best dominating target subgraph at runtime?
Q2 The embedding constraint has the interesting property that it is not invariant under rotations.  Do the authors have any observations about whether this results in difficulties in the learning?
Q3: In section 3.2, you comment "This benefit is a result of avoiding the loss of information when pooling node embeddings and a better inductive bias stemming from order embeddings."  Could you describe the evidence in support of this hypothesis?
Q4: Why is it that in Table 2, FastPFP performs better than NeuroMatch for synthetic data?  It's not a concern for the overall eval, but NeuroMatch by design seems as though it should perform well in this setting.

Other comments:
* I suggest introducing the set embeddings (McFee & Lanckriet '90) in the paper.

Summary of recommendation: I think this is a borderline paper.  The approach seems reasonable.  There is some novelty in the problem choice, in the framework of generating training data, and in the geometric constraints on the embeddings, but none of these alone is a significant advance in the area.  The empirical results look quite good, but of course the problem hasn't had the same level of scrutiny as better-known pieces of the ML canon.

---

> ### Author Response · Authors · 2020-11-24
> **Response to Reviewer #3**
>
> We thank the reviewer for their insightful feedback and for noting that the geometric embeddings are a "well chosen" approach and "perform well in practice," and our experiments are thorough, with "nice study of alternate sampling approaches."
>
> We emphasize that NeuroMatch is the first neural approach that combines advances of GNNs and geometric embeddings for the subgraph matching problem. An 18% improvement demonstrates that a neural approach is a feasible new direction that can stand up against other classes of approaches. We demonstrate that other state-of-the-art neural alternatives cannot be adapted to the subgraph matching task well, indicating that NeuroMatch successfully introduces new techniques to this harder but very important problem.
>
>
> 1. RE Q1. "Could the authors speak a little to efficiently querying for the best dominating target subgraph at runtime?"
>
> We note that the range tree data structure could provide a space and time efficient solution for comparing whether a query embedding satisfies the order embedding constraint relative to a set of target graphs. However, since embedding comparisons are very fast due to our approach (constant in graph size, millions of comparisons per second), we find that runtime already compares favorably with exact methods without this optimization. For example, constant-time embedding comparison already allows NeuroMatch to achieve high throughput in applications that require it, while exact methods often get stuck on a few hard examples (Appendix G1). Additionally, not constraining the method to a particular data structure allows us to flexibly define a finer-grained compatibility score (eq (4)) for higher accuracy predictions. We will discuss it in the final version.
>
> 2. RE Q2. "Do the authors have any observations about whether [lack of rotational invariance] results in difficulties in the learning?".
>
> We did not find that lack of rotational invariance led to difficulties in training. There is no fluctuation in performance once the training stabilizes. Generally the property of order invariance of graphs is already captured by the aggregation functions of GNNs. Unlike images, graphs have no notion of rotations, and thus invariance of embeddings with respect to rotation of data is not well-defined. Invariance of order constraint with respect to rotation of embedding is also not well-defined, since rotation is high-dimension GNN embeddings does not directly correspond to any operation on graphs.
>
> 3. RE Q3. Clarify and justify the claim, "This benefit is a result of avoiding the loss of information when pooling node embeddings and a better inductive bias stemming from order embeddings".
>
> Two major advantages of NeuroMatch are (1) its use of anchor nodes to break graph symmetries, increasing the expressive power of GNNs (elaborated in appendix H), and (2) its use of order embeddings to capture the inductive biases that the subgraph relation is transitive and anti-symmetric and has non-empty intersection (elaborated in appendix A). We will elaborate in the final version.
>
> 4. RE Q4. "Why is it that in Table 2, FastPFP performs better than NeuroMatch for synthetic data?"
>
> Indeed, it is interesting that NeuroMatch does not match the performance of either FastPFP or the MLP baseline on the synthetic dataset in Table 2, despite outperforming the MLP baseline in Table 1. The discrepancy in performance between the two tasks suggests future room to improve the aggregation strategy used to solve Problem 1. To be sure, we note that NeuroMatch has no particular a priori advantage in the synthetic case, since in Table 2, it is trained and tested on this dataset in the same way as the real-world datasets. (The motivation behind the synthetic data is for pretraining in later experiments, as shown in Appendix Table 6.)
>
> ### We further want to point out misunderstandings in the suggested weakness.
>
> 1. RE: The performance is similar for different sampling strategies
>
> We hypothesize that consistent high performance regardless of the training strategy is due to the diversity of graphs encountered in all search strategies. For example, the following are statistics of query graphs sampled with the random BFS and degree-weighted sampling strategies on the ENZYMES dataset. Both strategies cover a similar and wide range of graph statistics. We will include the analysis in the final version.
>
> * Enzymes: degree-weighted sampling
>
> n verts. Mean: 25.16. Std: 3.15.
>
> n edges. Mean: 46.36. Std: 8.18.
>
> radius. Mean: 4.57. Std: 1.15.
>
> clustering. Mean: 0.35. Std: 0.17.
>
> density. Mean: 0.16. Std: 0.03.
>
> * Enzymes: random BFS
>
> n verts. Mean: 17.50. Std: 6.19.
>
> n edges. Mean: 33.25. Std: 13.72.
>
> radius. Mean: 2.75. Std: 0.50.
>
> clustering. Mean: 0.32. Std: 0.16.
>
> density. Mean: 0.25. Std: 0.09.

---

### Official Review · AnonReviewer4 · 2020-10-29
**Several aspects poorly defined**

**Rating:** 3
**Confidence:** 5

**Review:**


This paper presents a method to solve the subgraph matching problem based on training a graph neural network to produce embeddings of 10-hop, apparently labelled, node neighborhoods in a way that allow for recognizing subgraph relationships among graphs via dominance relationships among their embeddings.

The results presented in the paper appear occasionally simply too good to be true, while several aspects of the presented method are poorly defined, or totally undefined. Among them the following:

1. Algorithm 1, which provides the main tool to be used in order to answer Problem 1, is defined as returning a subgraph of G_T that is isomorphic to G_Q. However, it is not clear how that is supposed to happen, and the algorithm's pseudocode provides no clue about it. Appendix D revisits this question, and offers that the Hungarian algorithm may be used in case when an explicit matching is desired, yet there is no elaboration on the topic and no results on it.

2. The pseudocode states that a binary prediction is made based on the *average* score of all value of subgraph prediction function f(z_q, z_u); there does not appear to be any way of returning an subgraph as output, apart from this prediction. In itself, the idea that merely taking an average of a binary subgraph prediction function over all node pairs would result in a correct prediction appears simply too good to be true.

3. The paper lacks a sufficient explanation of details regarding this predictive method. There appears to be a threshold t, or perhaps α (both names are used for it) on the magnitude of a violation E(z_q, z_u) on the subgraph constraint among the embedding dimensions of q and u. This threshold is mentioned in Section 2.3 by two names, and then it is never mentioned again. No discussion is offered on what values it has in experiments.

4. Appendix D mentions a sweep over hyperparameters, whose effect is supposed to be shown in Table 4. However, Table 4 shows the accuracy of matching on the ENZYMES data set in an unsystematic manner. No discussion is offered on the threshold parameter. The section on Hyperparameters lists a number of conclusions, which do not appear to relate directly to Table 4.

5. The writing fluctuates several times between the idea of using categorical node features and not using them. Eventually, the matter is left undefined, and the reader cannot tell whether such features are used or not. Section 2.1 leaves the question open. Appendix D comes back to this question, and defines that, with the FastPFP method, a prediction score uses the features matrices of nodes. This is the first time when it eventually becomes apparent that experiments are indeed using node labels all the way, yet the matter is not previously discussed.

6. The size of a k-hop subgraph is not explicitly specified. Section 2.1 suggests that k = 10 in experiments. The value of k is called number of layers. Table 4 in the Appendix also refers to a number of layers. It is not clear whether that refers to k or to a neural network architecture.

---

> ### Author Response · Authors · 2020-11-24
> **Response to Reviewer #4**
>
> We thank the reviewer for their valuable suggestions to improve the clarity of the paper. We note that the review does not raise concerns about the algorithm but rather its description and downstream usage, which we believe can be addressed by clarifying the writing and scope of our paper. We have made many improvements to the description of the algorithm and experiments accordingly.
>
> Firstly we want to point out that “simply too good to be true” is not an appropriate point of weakness. We have provided code in appendix for full reproducibility of the results. We further explain experimental setups and hyperparameters in detail in both text and appendix. Furthermore, we discussed limitations of this approach in paper and pointed out many future directions for this pioneering approach.
>
> * RE Q1. The "algorithm's pseudocode provides no clue about" finding an explicit matching.
>
> We emphasize that Problem 1 and 2 both explicitly focus on solving the decision problem of subgraph isomorphism (which is already NP-complete), hence returning an exact alignment is not the main goal of this work. We have clarified the wording of Algorithm 1 accordingly.
>
> * RE Q2. "There does not appear to be any way of returning an subgraph as output."
>
> Our work focuses on the decision problem of subgraph matching rather than finding an explicit matching (returning the matched subgraph as output). Since our method returns an alignment matrix, this matrix can be post-processed with an arbitrary alignment resolution algorithm in the same way as IsoRankN and FastPFP. However, our method itself is agnostic to such algorithms, hence we do not focus on these post-processing approaches -- they are orthogonal to the contributions of GNN+order embedding in this work. Furthemore, we do not believe our strategy is "too good to be true" in this case. If NeuroMatch can provide a perfect alignment matrix, this problem could be solved with a very simple strategy of testing whether any of the entries of the matrix is zero (indicates a match). Taking the mean of the matrix entries is simply an empirically better-performing ensemble strategy due to prediction errors of entries in the alignment matrix. We will clarify this fact in the paper.
>
> * RE Q3. There "lacks a sufficient explanation of details regarding" thresholds α and t.
>
> We point out a misconception: the reviewer appears to be conflating α and t, which are distinct concepts. α is a hyperparameter to the max margin loss which enforces, at training time, that two graphs in a negative example are separated by a margin. We found α=0.1 to perform well (although the performance is not sensitive to α). t is a learned threshold that makes the binary subgraph prediction at inference time. For ENZYMES, we find that the learned value of t is 0.301. We added a clarification of the role of α in the margin loss section of the revised paper.
>
> * RE Q4. In Appendix D, "No discussion is offered on the threshold parameter."
>
> The section on Hyperparameters lists a number of conclusions, which do not appear to relate directly to Table 4": We found early on that α=0.1 performs well on our task, but that performance is very robust to varying α. We added a clarification in the Hyperparameters section that we give additional tips about architectural choices, beyond those compared in Appendix Table 4, that we found helpful for implementation.
>
> * RE Q5. The use of node features "is left undefined".
>
> While the inclusion of node labels is already described in the problem definition and experiments (for datasets with features, "we require that the matching takes categorical features of nodes into account"), we have added additional clarification in the problem definition section that the model takes the node labels as input. Furthermore, we emphasize that NeuroMatch is the first neural approach to subgraph matching, and many variations such as the use of node labels are simple extensions of the foundational geometric embedding approach.
>
> * RE Q6. "It is not clear whether" the number of layers "refers to k or to a neural network architecture".
>
> We have already emphasized in the paper that the two notions of hop size and number of layers are equivalent: "By using a $k$-layer GNN to embed node $u$, we are essentially embedding/capturing the $k$-hop network neighborhood structure $G_u$ around the center node $u$." We have added a sentence to this paragraph making this notational convention explicit. Hence the number of layers in Appendix Table 4 indeed refers to k.

---

### Official Review · AnonReviewer2 · 2020-10-30
**A paper with novel idea. Above the acceptance threshold.**

**Rating:** 6
**Confidence:** 5

**Review:**

This paper proposed a new algorithm for performing subgraph matching under deep learning framework. To this end, the problem is formulated as a node-level metric learning problem. A main contribution of the paper is to introduce partial order preserving property for the metric and yields significant improvements over existing methods. I believe the paper is above the threshold in general.

However, I have the following concerns about the paper.

1) I see the author conduct the experiment with k up to 10. While this is a rational number for synthetic and small scale graphs, a feasible k can be much larger in other scenarios. I would like to see how the choice of k can be influenced by different datasets and what is the connection.

2) The result of the matching can rely heavily on the quality of the construction of the graphs. However, in real cases, a query graph may be constructed using some heuristic ways (e.g. k-nearest, \epsilon-ball or Delauney in graphics). This heuristic construction of graphs may lead to changes to the topology of the graph. Therefore, I think the authors should present more discussion on how to handle such uncertainty in practice.

3) The way of generating training and testing samples seems naive. In practice, both subgraphs of query and target can have semantic meanings, thus a random sampling of subgraphs cannot fully represent the true distribution. I would recommend the authors consider about more practical cases rather than synthetic ones.

4) What changes were made to the two selected baselines (GMNN and RDGCN) to adapt the subgraph matching tasks? It is necessary to give more details showing how the modifications are made. This can help to better understand the baselines.

5) While the authors claimed that subgraph isomorphism can be defined with both node and edges features, I didn't see any part of the paper handling edge features. Throughout the paper, only node features are considered. While edge features are essentials in a large variety of applications (e.g. drug compound matching, DNA matching), I think missing this portion can greatly limit the contribution of this paper. To me, it also seems that the proposed method cannot be readily extended to edge features. I suggest the authors to at least discuss the feasibility of extending this method to edge features.

I would consider raising the rating if the authors can address the aforementioned questions well.

---

> ### Author Response · Authors · 2020-11-24
> **Response to Reviewer #2**
>
> We thank the reviewer for their insightful review and for acknowledging the novelty and effectiveness of our approach. We emphasize that NeuroMatch introduces the first neural framework for subgraph matching, and many extensions (such as using the edge features) are natural and possible in the future work. In response to the reviewer's questions (numbered Q1-Q5):
>
> * RE Q1: "How the choice of k can be influenced by different datasets and what is the connection".
>
> Using a variety of query sampling strategies, we empirically found that a hop size of 10 is sufficient to cover a wide range of queries in real-world networks. We show in paper that the value of k only needs to be big enough to allow propagation of any node in a query to other nodes in the query. In practice, the diameter of the query is a good guideline for the value of k. Empirically, we ran ablations in Appendix Table 4 (SAGE 64-dim, dropout=0) to show that the model performance is stable but generally increases as k (number of layers) increases until it plateaus.
>
>
> * RE Q2: "How to handle...uncertainty" in the construction of query graphs?
>
> The method is agnostic to ways of query construction. We also note that the training and evaluation queries sampled with our method (3 different algorithms of query construction are already considered in paper) span a large space of possible graph statistics, and our hard negative examples make our method robust to query perturbations. We report graph statistics for query graphs sampled with random BFS (3 hops) below, showing that they cover a wide range.
>
>
> * RE Q3. "Consider about more practical cases rather than synthetic ones" in evaluation.
>
> We note that NeuroMatch outperforms baselines consistently over several real-world datasets and sampling methods. We conducted experiments to show that our model generalizes to queries that are not only unseen in training, but also sampled from a different sampling strategies (random BFS, random walk, degree-weighted sampling). Although domain-specific semantically meaningful queries are important in practice, there exist no public datasets with such queries. Furthermore, our work primarily proposes a general strategy that then applies to many possible types of queries.
>
>
> * RE Q4. "What changes were made to the two selected baselines (GMNN and RDGCN) to adapt the subgraph matching tasks?"
>
> The main change for GMNN and RDGCN is to adapt the objective to that of subgraph matching without any change in their GNN architecture. Instead of predicting true for 2 isomorphic graphs, we train the models to predict true for a pair (Q, G) iff Q is a subgraph of G. We will clarify it in the paper.
>
>
> * RE Q5. "I suggest the authors to at least discuss the feasibility of extending this method to edge features."
>
> To handle edge features, one can simply retrain with a GNN architecture that handles edge features, and sample training examples with edge features; the order embedding approach still applies. We note that in this paper we introduce a new first neural approach to subgraph matching and there are many possible extensions in addition to those explored in the paper (with or without node features, generalization to new sampling methods and domains). We further note that many standard benchmarks in chemistry and biology (COX2, ENZYMES) contain node features but no edge features [1], so our current model can already handle these common use cases.
>
>
> * MSRC_21
>
> n edges. Mean: 88.84. Std: 30.46.
>
> clustering. Mean: 0.54. Std: 0.05.
>
> density. Mean: 0.14. Std: 0.04.
>
>
> * DD
>
> n edges. Mean: 69.18. Std: 68.10.
>
> clustering. Mean: 0.54. Std: 0.09.
>
> density. Mean: 0.19. Std: 0.09.
>
>
> * ENZYMES
>
> n edges. Mean: 26.46. Std: 12.26.
>
> clustering. Mean: 0.42. Std: 0.21.
>
> density. Mean: 0.29. Std: 0.11.
>
>
> [1] https://ls11-www.cs.tu-dortmund.de/staff/morris/graphkerneldatasets

---

### Decision · Program_Chairs · 2021-01-07
**Final Decision**

**Decision:**

Reject

**Comment:**

This paper proposes an interesting approach for learning to decide whether a query graph is isomorphic to a subgraph within the target graph.  The approach has a number of interesting aspects from the machine learning perspective, e.g. the anchored graphs and the order embeddings.  Empirical results show promise in ablation studies and against a few baselines.

However this paper also has a number of issues as pointed out by the reviewers, placing it right on the borderline.  Most notably the clarity of the presentation could be improved as it seems to confuse a few reviewers at various points.  Another thing that I’d like to highlight is that the way to convert the pairwise scores f(z_q, z_u) into the final decision about G_T and G_Q seems worthy of a longer discussion.  Is a simple average across all pairs the best we can do?  I imagine if the query graph is small but the target graph is large then even if the G_Q does match a subgraph of G_T the average score can be quite low.

Overall I do like the ideas proposed in this paper, but also recognize that the paper can benefit from more improvement, so I’d like to recommend rejection but encourage the authors to submit again in the next round.